# Development and characterization of a 2D porcine colonic organoid model for studying intestinal physiology and barrier function

Masina Plenge[1]*, Nadine Schnepel[1], Mathias Müsken[2], Judith Rohde[3], Ralph Goethe[3], Gerhard Breves[1], Gemma Mazzuoli-Weber[1], Pascal Benz[1]

1 Institute for Physiology and Cell Biology, University of Veterinary Medicine Hannover, Hannover, Germany, 2 Central Facility for Microscopy, Helmholtz Centre for Infection Research, Braunschweig, Germany, 3 Institute of Microbiology, University of Veterinary Medicine Hannover, Hannover, Germany

* Masina.plenge@tiho-hannover.de

## Abstract

The porcine colon epithelium plays a crucial role in nutrient absorption, ion transport, and barrier function. However ethical concerns necessitate the development of alternatives to animal models for its study. The objective of this study was to develop and characterize a two-dimensional (2D) *in vitro* model of porcine colonic organoids that closely mimics native colon tissue, thereby supporting in vitro research in gastrointestinal physiology, pathology, and pharmacology. Porcine colonic crypts were isolated and cultured in three-dimensional (3D) organoid systems, which were subsequently disaggregated to form 2D monolayers on transwell inserts. The integrity of the monolayers was evaluated through the measurement of transepithelial electrical resistance (TEER) and electron microscopy. The functional prerequisites of the model were evaluated through the measurement of the mRNA expression of key ion channels and transporters, using quantitative RT-PCR. Ussing chamber experiments were performed to verify physiological activity. The 2D monolayer displayed robust TEER values and retained structural characteristics, including microvilli and mucus-secreting goblet cells, comparable to those observed in native colon tissue. Gene expression analysis revealed no significant differences between the 2D organoid model and native tissue with regard to critical transporters. Ussing chamber experiments demonstrated physiological responses that were consistent with those observed in native colonic tissue. In conclusion, 2D porcine colonic organoid model can be recommended as an accurate representation of the physiological and functional attributes of the native colon epithelium. This model offers a valuable tool for investigating intestinal barrier properties, ion transport, and the pathophysiology of gastrointestinal diseases, while adhering to the 3R principles.

**Data availability statement:** All relevant data are within the manuscript and its Supporting information files.

**Funding:** The Federal Ministry of Food and Agriculture (BLE #28N-2-071-00) funded this work. The funders had no role in study design, data collection and analysis, decision to publish, or preparation of the manuscript.

**Competing interests:** The authors have declared that no competing interests exist.

## Introduction

The gastrointestinal tract (GIT) is vital for nutrient digestion and absorption, waste excretion, and metabolic homeostasis [1]. The GIT epithelium acts as a selective barrier, regulating the passage of nutrients, ions, and water. Tight junction proteins, such as occludin (OCLN), claudin (CLDN) and zonula occludens (ZO-1), are crucial in maintaining the integrity of the epithelial barrier by providing both a "fence" and a "gate" or barrier function [2,3]. The fence function serves to separate the apical from the basolateral sides of the epithelial cells, ensuring the proper distribution of membrane proteins and lipids. Moreover, the mucus layer, which is composed of mucins (MUC) and is secreted by goblet cells, serves as the primary defence against luminal pathogens and mechanical stress [4]. Various transporters, channels, and carriers facilitate the transport across the epithelium, such as the epithelial sodium channel (ENaC), sodium-hydrogen exchanger (NHE), calcium-dependent chloride channel (CaCC), and cystic fibrosis transmembrane conductance regulator (CFTR). Each of these regulates the transport of specific ions and molecules. The function of the GIT has historically been studied by animal experiments, with pigs being commonly used as models due to their relevance to both, pig-specific diseases and human intestinal conditions [5].

However, with the increasing relevance of ethical concerns and the principles of replacement, reduction, and refinement (3R) [6], alternative methodologies such as cell culture-based systems are gaining attraction. Cell lines have been instrumental in unravelling certain aspects of GIT physiology. Nevertheless, the usefulness of these models has often been limited by their focus on singular cell types. This limitation highlights the need for more comprehensive models that encompass the diverse cell populations found within the gastrointestinal epithelium. The colon hosts a variety of cell types, including enterocytes, goblet cells, enteroendocrine cells, tuft cells, and stem cells, each contributing uniquely to its function [7,8]. To overcome this limitation of a single cell type, the organoid model was introduced in 2009 [9], revolutionising our ability to model the complex architecture and functionality of the intestinal epithelium. Organoids are derived from stem cells, which impart two crucial properties to them. Firstly, these cells can differentiate into all cell types of the epithelium from which they were obtained. Secondly, organoids possess the ability to self-renewal [10]. Organoids cultivated in a 2D-monolayer system are highly valued as effective tools for studying barrier properties for drug discovery and development [11]. According to Hoffmann et al. [12], porcine jejunum organoid cultures are a useful model for investigating physiological transport properties of the epithelium. Nevertheless, a comprehensive model that takes into account the physiological attributes of the porcine colon is still lacking. Colon organoids offer unique advantages for investigating colon-specific disease mechanisms and treatments. They effectively model the thicker mucus layer that is essential for maintaining epithelial barrier integrity, and facilitate research into critical functions such as water and electrolyte absorption. Thus, this study aimed to delineate the physiological functionality of a 2D culture of porcine colonic organoids, emphasizing gene expression relative to native tissue and elucidating the transport properties of the organoid-derived epithelium.

## Materials and methods

For this protocol, two healthy pigs were sacrificed by captive bolt shooting and exsanguination. According to the Animal Welfare Act (directive 2010/63 EU), this (slaughter and removal of tissues) is not classified as an animal experiment, but must be reported to the University Animal Welfare Officer (registration number TiHo-T-2017–22 and TiHo-T-2024–5).

### Culture of 3D-organoids

To generate 3D organoids, intestinal crypts were isolated from the porcine colon. The cultivation of 3D organoids has already been described in detail by Hoffmann et al. [12]. In this instance, a 10 cm segment of the proximal porcine colon was employed for the isolation of crypts. Briefly, the organoids were cultured in Matrigel (Corning®, Kaiserslautern, Germany; Matrigel® Basement Membrane Matrix) droplets of 50 µl in a 24-well plate. Every 2–3 days, the organoid medium (S1 Table) was changed and at least once a week the droplet was resolved and the 3D organoids were passaged. The organoid medium was then aspirated, ice-cold PBS was added, and the Matrigel droplet dissolved by pipetting. The suspension was collected in a tube and was centrifuged at 500 x $g$ for 10 minutes at 4 °C. The pellet was resuspended in medium and diluted 1:5 with Matrigel. The plate was incubated at 37 °C for at least 30 min and overlaid with 500 µl of organoid medium per well.

### Culture of 2D-organoids

2D organoids were generated by enzymatic and mechanical disintegration of 3D organoids as described in the following. The medium was aspirated and the Matrigel droplets were resolved with 1 ml ice cold PBS, at this stage 3–4 wells were pooled. The solution was transferred to a reaction tube with 10 ml ice cold PBS and then centrifuged at 250 x $g$ for 10 min at 4 °C. The supernatant was removed, and the pellet was resuspended in 1 ml 0.05% Trypsin/ EDTA (Gibco™, Thermo Fisher) and incubated for 5 min at 37 °C. Afterwards, the solution was resuspended 20 times with a 1,000 µl tip and 15 times with a 200 µl tip, which was mounted on a 1,000 µl tip, all steps being performed on ice. 10 ml Advanced DMEM/F-12 (Thermo Fisher scientific, Waltham, USA) supplemented with 10% (v/v) fetal bovine serum (FBS; Sigma-Aldrich, Darmstadt, Germany; catalogue no.: F7524) were added to the cell suspension and centrifuged at 1,000 x $g$ for 10 min at 4 °C. After removing the supernatant, the pellet was resuspended in 1 ml monolayer medium (S2 Table). Cells were counted and $1.5 \cdot 10^5$ cells were seeded on cell culture inserts (Corning®; Snapwells® or GREINER BIO ONE; ThinCert® Cell Culture Insert 12 Well; diameter: 12 mm; pore size: 0.4 µm) pre-coated with 1:40 (v/v) Matrigel in PBS. Every 2 days the monolayer medium was replaced and the transepithelial electrical resistance (TEER) was measured with a STX4 electrode (EVOM³; World Precision Instruments, Berlin, Germany). The differentiation was started at cultivation day 8 by changing the medium to differentiation medium (S3 Table). Differentiation medium was replaced daily and the TEER measured. Experiments were conducted on the 10th day of cultivation.

### RNA isolation and reverse transcription

On the 10th day of cell cultivation, the cells were harvested for analysis. After aspirating the medium, cells grown on transwells were washed with ice-cold PBS and the cellular layer was detached from the membrane with a spatula. For the examination of tight junction proteins, cells were also collected on the 4th, 7th, and 9th days of cultivation. Samples were centrifuged at 1,000 x $g$ for 10 minutes at 4 °C, followed by swift freezing in liquid nitrogen and subsequent storage at -80 °C. For comparison, native tissues from the same breed, age and colon section as the organoids donors was used. These tissues were obtained from banked samples. The RNA extraction was performed with the RNeasy Plus Mini Kit (Qiagen, Hilden, Germany) according to the manufacturer's instructions. The concentration of the isolated RNA was determined by spectrophotometric measurements using the NanoDrop™ One (Thermo Scientific™). The cDNA synthesis was performed via reverse transcription using TaqMan™ Reverse Transcription Reagents Kit (Applied Biosystems, Roche Molecular System, Darmstadt, Germany).

## Quantitative real-time PCR

To determine mRNA expression of the target genes, quantitative realtime PCR was used. The gene expression of the genes in S4 Table was determined using the SYBR Green® PCR assay as previously described by Elfers et al. [13]. For the targeted gene *MUC4* the annealing temperature was 63 °C and for *ENaC alpha* and *NHE3* the annealing temperature was 65 °C for all other genes the annealing temperature was 60 °C. The gene expression of the target genes (S5 Table) were determined by TaqMan®. The reaction mixture contained TaqMan™ Gene Expression Master Mix (Applied Biosystems), 16 ng reverse transcribed cDNA, 300 nM of the specific primers and 100 nM of specific probe at a total volume of 20 µl. For each gene, a negative control and a standard series, which is a dilution series of the amplification product, were used to determine the primer efficiency. The amplification and detection was performed with the real-time PCR cycler Bio-Rad CFX96™ (BIO-RAD Laboratories, INC., Hercules, USA). The housekeeping genes of the 60S ribosomal protein L4 (RPL4) and the 40S ribosomal protein S18 (RPS18) were used to normalise the gene expression of the genes of interest. The relative gene expression was calculated with the Pfaffl [14] method and the mean of the two houskeeping genes. For the gene expression analysis, RNA was isolated from four independent 2D cultures of organoids, with each sample having two technical replicates to ensure the reliability and consistency of the data.

## Ussing chamber experiments – transport physiology

In order to investigate epithelial transport characteristics, 2D monolayers were mounted in Ussing chambers at the end of differentiation as described by Hoffmann et al. [12]. After an initial period of 5 min, the tissues were set to short-circuit-conditions for another 15 min to allow equilibration. The short-circuit current ($I_{sc}$) and the epithelial membrane resistance (Rt) were monitored every 6 seconds for the whole experiment. The 2D culture was than challenged with one of the substances stated in Table 1. Each substance was tested individually in a chamber, followed by treatment with forskolin for the purpose of assessing viability. In the case of aldosterone, the 2D culture underwent a 3 h, 8 h and 48 h preincubation with aldosterone before Ussing chamber experiments were performed. The epithelia, which has been preincubated with aldosterone, and the control, which had not been preincubated with aldosterone, were subsequently treated in the Ussing chamber with 100 µM amiloride. Except for forskolin and aldosterone dissolved in DMSO, all other substances are diluted in *aqua destillata*. Serosal addition of mannitol after the addition of glucose was carried out to ensure osmotic stability. For analysis, the basal values of $I_{sc}$ determined in the Ussing chamber experiments were the mean calculated from the last ten values prior to the addition of an additive. The changes in short-circuit currents ($\Delta I_{sc}$) are the difference between the maximum or minimum value after the addition of each substance and the the basal value. Each Ussing chamber setup was repeated in four independent experiments, with three technical replicates per condition, ensuring the reproducibility of the findings.

**Table 1. Application for Ussing chamber experiments (all chemicals were obtained from Sigma-Aldrich, Darmstadt, Germany).**

| Substance | Concentration | application | Incubation time |
|---|---|---|---|
| DMSO | 1:1000 | serosal | 15 min |
| glucose<br>mannitol | 10 mM<br>10 mM | mucosal<br>serosal | 15 min |
| carbachol | 10 µM | serosal | 15 min |
| forskolin | 10 µM | serosal | 15 min |
| amiloride | 50 µM<br>100 µM<br>500 µM<br>1 mM | mucosal | 15 min |
| aldosterone | 3 nM,<br>1 nM,<br>0.3 nM | serosal | Preincubation for 3 h, 8 h or 48 h |

## Scanning electron microscopy

Electron microscopy sample preparation was performed in a similar way as previously described [15]. In brief, cells on transwells were fixed in a 0.1 M EM Hepes buffer with 5% formaldehyde and 2% glutaraldehyde and washed twice to remove aldehyde residues. Dehydration was carried out in a gradient series of EtOH on ice (10%, 30%, 50%, 70%, 90%), each step for 10 min and two steps (100% EtOH) at room temperature. Ethanol was used to prevent damage of the transwell membrane. Afterwards, critical point drying (CPD300 from Leica) and sputter coating with gold palladium (SCD 500 from Bal-Tec) was performed. Samples were visualized with a field-emission scanning electron microscope Merlin (Zeiss) at an acceleration voltage of 5 kV and making use of both, an Everhart Thornley HESE2 and an inlens SE detector.

## Data analysis and statistics

TEER values of the 8th day of cultivation were set to 100% and the others calculated accordingly. The statistical analyses were performed using Prism version 9.0.0 (GraphPad, San Diego, USA). The normal distribution of all data was tested with the D'Agostino & Pearson test. Comparison between mean values of the gene expression on the 10th cultivation day and native colon tissue was conducted using the unpaired t-test. The one-way ANOVA was employed to compare the gene expression on various cultivation times followed by Tukey's test for post-hoc analysis to correct for multiple comparisons. The paired t-test was employed to compare the mean basal and maximum/minimum $I_{sc}$ values in the presence of a normal distribution. When the data did not follow a normal distribution, the Wilcoxon test was used for the same purpose. A two-way ANOVA was used to evaluate the effects of different cultivation times and aldosterone concentrations. Basal and minimum $I_{sc}$ values were compared as part of the analysis. Post-hoc analysis was performed using Dunnett's test to account for multiple comparisons. Differences were considered statistically significant when p value <0.05. Within figure legends, the number of independent experiments is indicated by the symbol "n."

## Results

### TEER progression shows robust cellular monolayer integrity

Measurement of TEER provides an indication of the integrity of the cellular layer. Monolayers were confluent on day 8 of cultivation and the differentiation was started by changing the cultivation medium from monolayer medium to differentiation medium. Therefore, the electric resistance was set to 100% on this day, and subsequent TEER measurments were calculated relative to this baseline value. The progression of the TEER values showed an increase until the 8th day of cultivation and formed a plateau during the differentiation period (Fig 1).

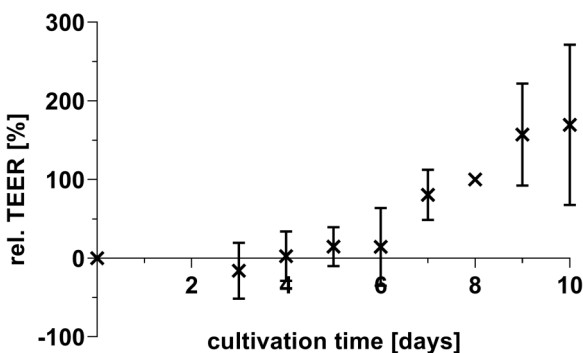

**Fig 1. Change of the transepithelial electrical resistance (TEER) during the 2D cultivation of the porcine colonic organoids.** The organoids were cultivated on transwell inserts (0.4 μm pore diameter). On the 8th day of cultivation the mean TEER value was 187.4 ± 28.3 Ω·cm² (mean ± SEM). Displayed is the mean ± SD, n = 4.

### Scanning electron microscopy reveals different cell types

Electron microscopy revealed an intact monolayer of 2D organoids at day 10 of culture. The monolayer was characterized by the development of well-defined microvilli at the apical surface. (Fig 2A, B). The goblet cells are indentified by multiple orifices, which arise from the fusion of mucin-containing granules (Fig 2A, B). This process has resulted in the formation of a mucus layer that covers the epithelium (Fig 2B).

### RT-PCR demonstrates gene expression concordance between 2D organoids and native tissue

The gene expression of target genes was determined in 2D cultures of organoids and compared with gene expression in native porcine colon tissue. The focus was on key genes that are central to intestinal function, particularly those that control absorption and secretion and are critical for optimal colonic function. Genes involved in the transport of $Na^+$, including ENaC and NHE, exhibited no differential expression in the organoid model when compared to native tissue. This pattern was similarly observed for genes for $Cl^-$ secretion (CFTR and CaCC), and the gene encoding for the uptake of glucose (SGLT1). Moreover, the expression of the mucin genes (MUC) demonstrated no significant difference between the 2D organoid culture and native tissue (Fig 3).

### Gene expression of tight junction proteins during cultivation is constitutive

The development of tight junction protein gene expression was examined on 4th, 7th, 9th and 10th day of cultivation. CLDN3 showed no difference in gene expression during the cultivation period on 4th, 7th, 9th and 10th day. The same was true for CLDN2 with a higher variance on 4th day. The other transmembrane tight junction protein OCLN also showed no significant differences over the cultivation period, but a slightly higher expression on the last day of cultivation (10th day). The tight junction associated protein ZO-1 showed no differences in gene expression over the cultivation period and compared to native colonic tissue (Fig 4).

### Ussing chamber studies show physiological transport capacity of 2D-organoids

The addition of DMSO, glucose, forskolin and carbachol was used to alter the $I_{sc}$ in Ussing chamber experiments (Fig 5). The addition of DMSO (0.1%), which was used as a solvent control, showed no effect to the $I_{sc}$. The same was observed after the addition of carbachol ($10^{-5}$ mol·l$^{-1}$), which stimulates the $Ca^{2+}$- dependent $Cl^-$ secretion. A significant increase in $I_{sc}$ was followed after the addition of glucose ($10^{-2}$ mol·l$^{-1}$), which was added to test the functionality of the SGLT1, which

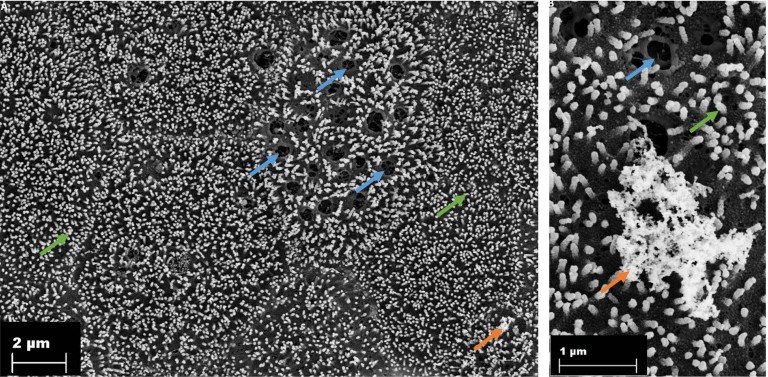

**Fig 2. Visualization of microvilli, goblet cells and mucus in the 2D culture of porcine colonic organoids.** Green arrows: microvilli; blue arrows: mucin-containing granules; orange arrows: mucus.

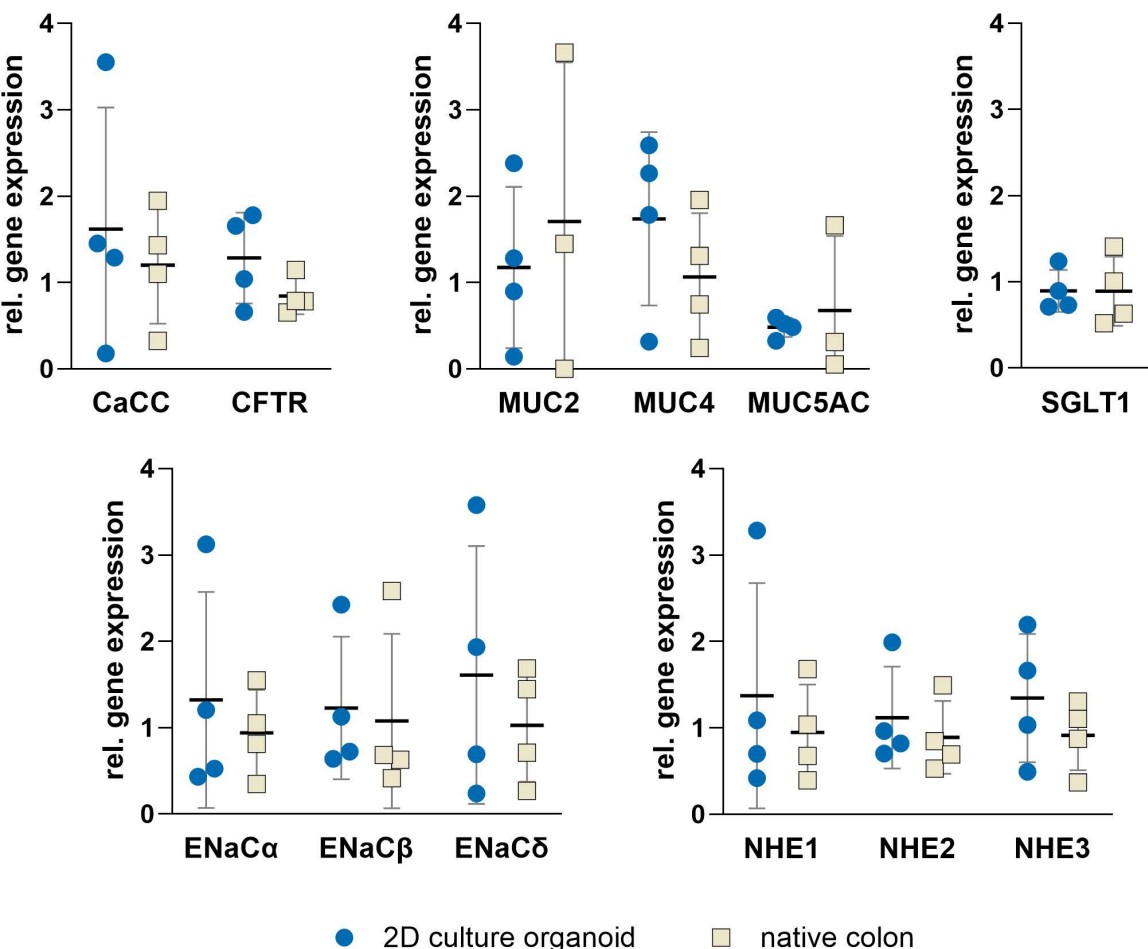

**Fig 3. Gene expression of calcium dependent chloride channel (CaCC), cystic fibrosis transmembrane conductance regulator (CFTR), the domains of epithelial sodium channel (ENaC α, ENaC β, ENaC γ, ENaC δ), mucin 2, 4 and 5AC (MUC2, MUC4, MUC5AC), the sodium-potassium-exchanger (NHE1, NHE2, NHE3) and sodium/glucose cotransporter 1 (SGLT1) in 2D cultures of porcine colonic organoids in comparison to native porcine tissue.** Ribosomal protein L4 (RPL4) and ribosomal protein S18 (RPS18) were used as reference genes and for normalization. Values shown: mean±SD, n=4, unpaired t-test was performed with no significant differences.

cotransports Na⁺ with glucose. Forskolin ($10^{-5}$ mol·l⁻¹) activates the cAMP-dependent Cl⁻ secretion through CFTR resulting in a significant increase in $I_{sc}$ after serosal addition (Fig 5).

Amiloride influences the epithelial sodium channel in a dose-dependent manner leading to a significant decrease at 100 µM and 500 µM (Fig 6A). This effect could not be observed with 50 µM or 1 mM amiloride. The addition of forskolin after amiloride led to a significant increase in $I_{sc}$, but showed no concentration-dependent effects (Fig 6B).

After the preincubation period with aldosterone, which activates apical Na⁺ channels, the monolayer was challenged with amiloride in the Ussing chamber. The basal $I_{sc}$ did not undergo a notable alteration when 0.3 nM aldosterone was present. When the two higher concentration were introduced, the dispersion of the basal values increased, yet this did not result in a significant rise in basal $I_{sc}$ (Fig 7). Notably, incubation with 1 nM aldosterone resulted in the smallest reduction in $\Delta I_{sc}$. Preincubation showed significant concentration effects (Fig 8). Conversely, incubation with 0.3 nM aldosterone did not yield a significant change in $\Delta I_{sc}$ for the 3, 8, or 48-hour preincubation periods (Fig 8). Significant alterations, however, were observed for the concentration and incubation time.

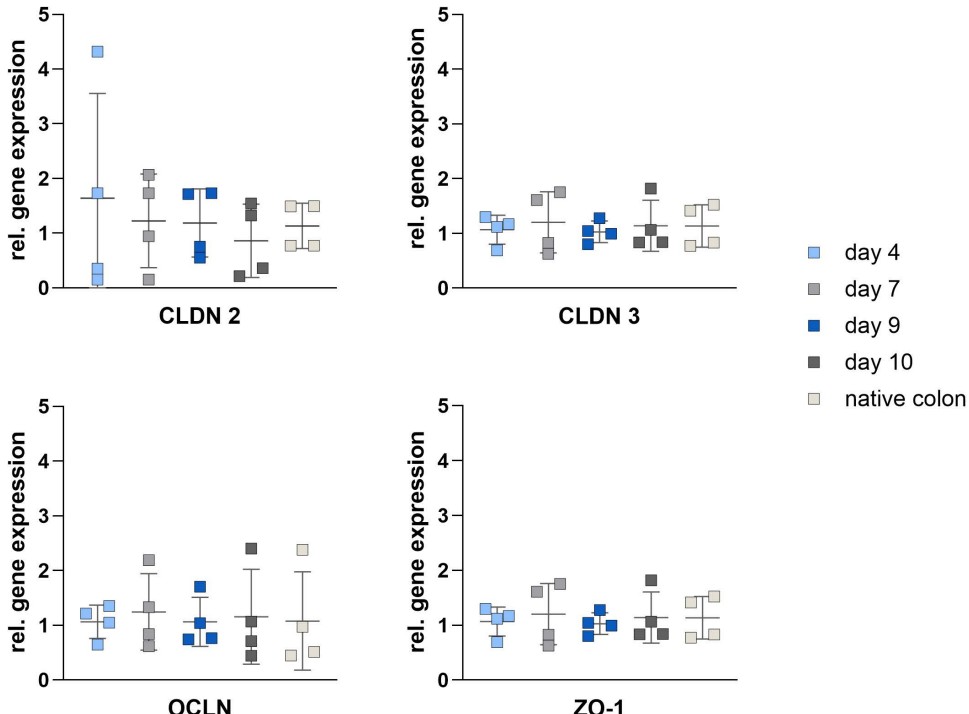

**Fig 4. Relative gene expression of the tight junction proteins claudin 2 (CLDN 2), claudin 3 (CLDN 3), occludin (OCLN) and zonula occludens 1 (ZO-1) in 2D-culture of porcine colonic organoids on different cultivation days (day 4, day 7, day 9 and day 10) and native porcine colon.** The reference genes ribosomal protein L4 (RPL4) and ribosomal protein S18 (RPS18) were used to normalize the expression. Values shown: mean±SD, n=4, One-way ANOVA was performed as a statistical test with no significant differences.

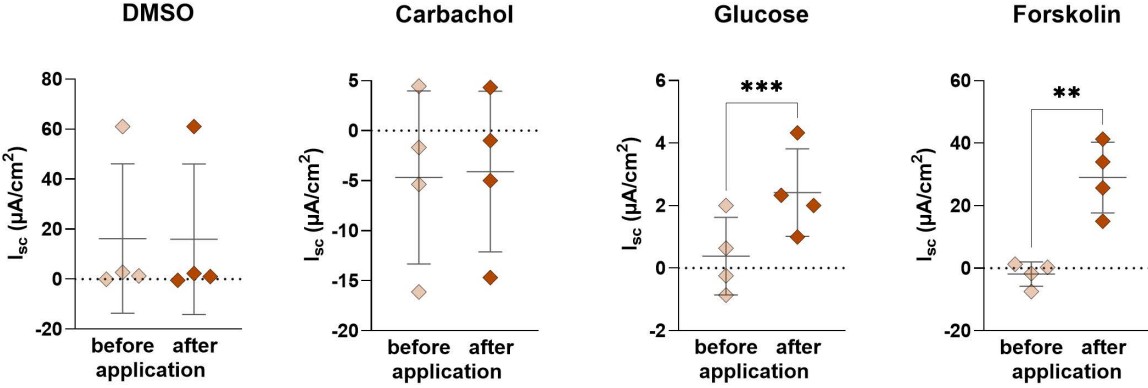

**Fig 5. Basal and maximal $I_{sc}$ values of 2D culture of organoids after addition of DMSO, glucose, forskolin and carbachol using Ussing chamber experiments.** Values shown: mean±SD, n=4. For DMSO a Wilcoxon test and for the other a paired t-test was performed: **p<0.01, ***p<0.001.

## Discussion

The colonic epithelium serves as a critical barrier, regulating nutrient absorption and ion transport while protecting against pathogens and mechanical stress. This objective of this study was to develop a 2D organoid model that mimics the native porcine colon, therby facilitating detailed investigations of its functional properties. By assessing gene expression, tight

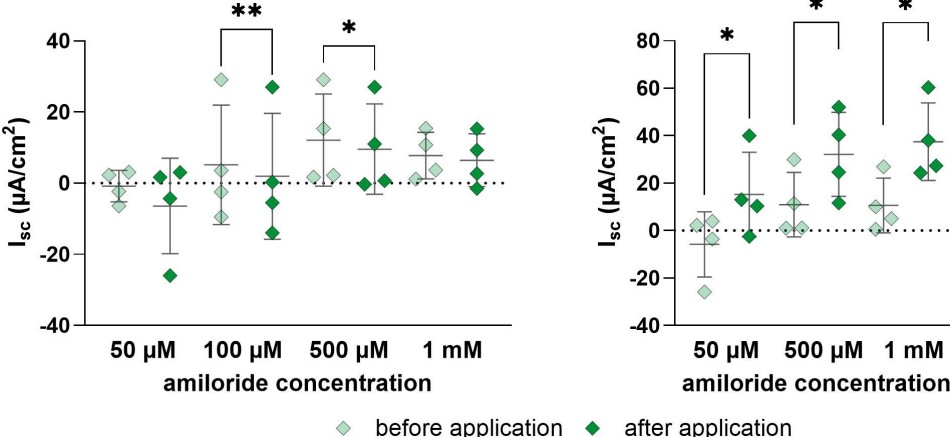

**Fig 6. Comparison of the I$_{sc}$ in 2D culture of colonic organoids with different concentrations of amiloride.** A: Basal and minimum I$_{sc}$ values after addition of 50 µM, 100 µM, 500 µM and 1 mM amiloride. B: Effect of 10 µM forskolin after the addition of amiloride. The forskolin was added 15 min after amiloride to the chamber. Values shown: mean ± SD, n = 4, paired t-test. * p < 0.05, ** p < 0.01.

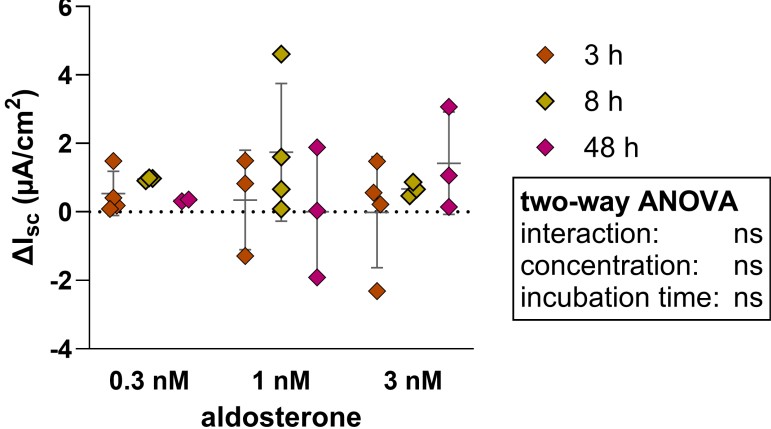

**Fig 7. Comparison of the basal I$_{sc}$ in 2D culture of colonic organoids, which were incubated with 0.3 nM, 1 nM or 3 nM aldosterone for 3 h, 8 h or 48 h, while the control was not incubated with any aldosterone.** The ΔI$_{sc}$ is the difference between of the aldosterone incubated basal I$_{sc}$ and the control basal I$_{sc}$. Values shown: mean ± SD of four independent experiments, two-way ANOVA with no significant changes.

junction formation, and transporter functionality, this model provides insights into epithelial barrier function and ion transport mechanisms. Furthermore, comparisons with native tissue revealed the model's capabilities and limitations, establishing it as a valuable platform for investigating colonic physiology, pathophysiology, and pharmacological interventions. Additionally, the model supports the 3R principles by reducing reliance on animal studies.

The colonic epithelium maintains selective barrier function, regulating the passage of nutrients, ions and water while safeguarding against luminal pathogens and mechanical stress. Electron microscopy imaging revealed the presence of microvilli on the polarised monolayer and identified goblet cells secreting mucus by exocytosis. These features are consistent with those of the colonic epithelium [16,17]. Hoffmann et al. [12], identified two distinct cell types in porcine colonic organoids: goblet cells and enterocytes. These findings align with the current study, further validating the cellular composition of the model.

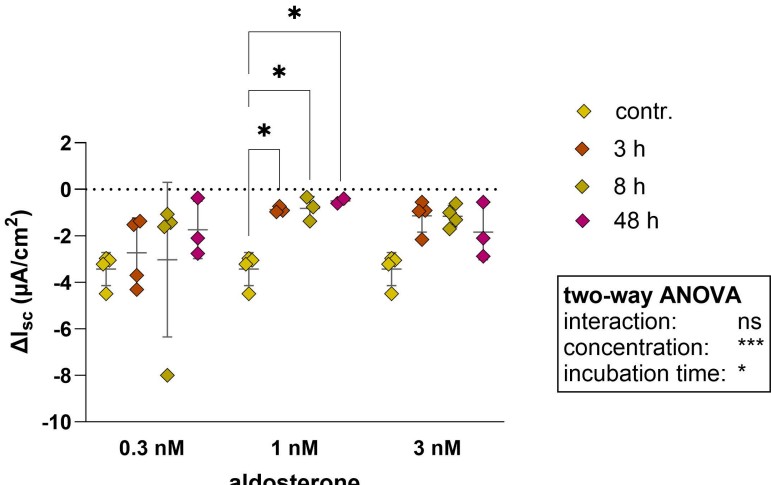

**Fig 8. Comparison of the** $\Delta I_{sc}$ **in 2D culture of colonic organoids, which were incubated with 0.3 nM, 1 nM or 3 nM aldosterone for 3 h, 8 h or 48 h, while the control (contr.) was not incubated with any aldosterone.** The $\Delta I_{sc}$ were calculated from the basal $I_{sc}$ and the minimal $I_{sc}$ after the addition of 100 µM amiloride. Values shown: mean ± SD, n = 4, two-way ANOVA. ns > 0.05, * p < 0.05, *** p < 0.001.

Mucins, the main structural components of mucus, were expressed in a manner consistent with the native colon. Specifically, MUC2 and MUC4 were observed, while MUC5AC, a mucin more common in the respiratory tract, was expressed at lower levels [18,19]. The similarity to native tissue underscores the physiological relevance of the model. Future studies could investigate mucin composition at the protein level to enhance the understanding of the functionalty and the role of mucus in diseases.

The barrier function is mediated by tight junction proteins, such as claudins (e.g., CLDN2, CLDN3), OCLN and ZO (e.g., ZO-1). The expression of claudins is organ-specific, and in the colon epithelium *CLDN2*, *CLDN3*, *OCLN*, and *ZO-1* are expressed [2,20,21]. No significant changes in the gene expression of these proteins were observed during the cultivation period. The formation of tight junctions between the cells results in an increase in TEER values, which can be observed during the formation of the monolayer in two-dimensional (2D) culture. Since the gene expression of the observed tight junction proteins remains unchanged, we assume that the tight junction proteins are formed constitutively during cultivation and are assembled as a tight junction when the cells grow together to form a monolayer. In 2D cultured organoids, TEER values increased until day 8, after which they stabalised, maintiaing this plateau until the end of the cultivation period.

Ion and nutrient transport mechanisms are vital for colonic functions. The expression of key transporters and channels, including *CaCC*, *CFTR*, *ENaC*, and *SGLT1*, mirrored native tissue. *SGLT1*, known to be expressed to a major extent in the small intestine, has been shown to be expressed in the large intestine in various animal species [22–24]. Functional studies using Ussing chambers revelaed a response to glucose via SGLT1, in contrast to native tissue, where no glucose response is observed[25]. It is possible that the discrepancy between the organoids and native tissue was due to the former cultivation with glucose as the only carbon source. Yoshikawa et al. [24] observed an increase in *SGLT1* expression in the colon of germ-free mice, where SCFA were absent as a carbon source. In contrast to the native colon, which primarily utilises SCFA derived from fermentation of fietary fibre for energy production [26], this experimental system relied exclusively on glucose. Future adaptations could incorporate SCFA in the culture medium to better reflect in vivo conditions.

The secretion of chloride through the CFTR, stimulated by forskolin, was observed to be consistent between the 2D organoids and the native tissue. Forskolin elevates intracellular cAMP levels, which subsequently activates CFTR, resulting inchloride secretion an and increased $I_{sc}$ [27,28]. Carbachol typically elevates intracellular $Ca^{2+}$ concentrations, which

are known to stimulate the CaCC-mediated Cl⁻ secration [29,30]. However, the lack of response to carbachol indicates an absence of functional CaCC [27,31], potenially due to differences in protein expression. Although no significant difference in gene expression between 2D organoids and native colonic tissue has been detected, the discrepancy between gene expression and protein functionality may be attributable to post-translational modifications, improper protein localisation, or the detection method's limited sensitivity. To elucidate this further, it would be beneficial to analyse protein expression in future studies using techniques such as Western blotting or immunohistology, which could provide insights into the presence and functionality of the CaCC protein.

However, the trend of lower responsiveness to carbachol is consistent with previous observations in porcine gastrointestinal tissues, which is also seen in porcine jejunum organoids [12,27].

Amiloride inhibition of the ENaC, which results in a reduction in $I_{sc}$ when measured in Ussing chambers, was dose dependent [32–34]. Higher concentrations potentially target NHE instead of ENaC, which align with prevoius studies [33,35]. The NHE is an electroneutral exchanger, a characteristic that prevents its detection using the Ussing chamber technique. In particular, it has been demonstrated that 1 mM amiloride inhibits the activity of NHE [36,37]. A concentration of 1 mM amiloride did not result in a reduction in $I_{sc}$ in the 2D culture of porcine colonic organoids, as the inhibition was not of the ENaC, but rather of the NHE. However, aldosterone-induced upregulation of ENaC, observed in native tissue, was not replicated [38–40]. Following the addition of the antagonist amiloride, a smaller decrease in $I_{sc}$ was observed than in the control, which was not incubated with aldosterone and showed a greater decrease in $I_{sc}$. This suggests that the 2D culture may lack certain regulatory mechanisms or timeframes requiered for a aldosterone mediated ENaC modulation.

The developed model exhibited robust barrier function, transporter expression, and physiological relevance to the native porcine colon. However, it also demonstrated notable limitations. The absence of functional CaCC and aldosterone responsiveness highlights areas for further refinement. Additionally, while gene expression aligned with native tissue, protein-level analyses are needed to fully validate the model. The reliance on glucose as the carbon source underscores the need for culture conditions that better mimic native metabolic environments, such as incorporating SCFA.

A limitation of this study is the small number of biological donors. The 3D organoids were derived from two pigs and expanded through multiple passages, a method commonly used in immortalized cell lines, which allows reproducibility. This approach maintained consistency across experiments despite the limited biological donors. Similar procedure have been employed in previous studies, such as the work by van der Hee et al. [41], who developed a standardized 2D monolayer system from porcine intestinal organoids, also employing two donor pigs. Increasing the number of biological donors in future studies would strengthen the generalization of findings. Nevertheless, this approach provides valuable insights and a strong foundation for further research.

This two-dimensional organoid model offers a promising platform for the study of porcine colonic epithelium, exhibiting a high degree of fidelity in terms of gene expression and functional properties. The model is applicable to research in epithelial physiology, infection studies, and pharmacological research, in alignment with the 3R principles. While the model has certain limitations, it provides a foundation for future studies aimed at refining and expanding its applicability.

## Supporting information

**S1 Table. Compostion of organoid medium.**
(DOCX)

**S2 Table. Composition of monolayer medium.**
(DOCX)

**S3 Table. Composition of differentiation medium.**
(DOCX)

**S4 Table. SYBR Green primers for SYBR Green® assay.**
(DOCX)

**S5 Table. TaqMan primers for TaqMan® assay.**
(DOCX)

## Author contributions

**Conceptualization:** Judith Rohde, Ralph Goethe, Gerhard Breves, Gemma Mazzuoli-Weber, Pascal Benz.

**Data curation:** Masina Plenge.

**Formal analysis:** Masina Plenge.

**Funding acquisition:** Judith Rohde, Ralph Goethe, Gemma Mazzuoli-Weber, Pascal Benz.

**Investigation:** Masina Plenge.

**Methodology:** Masina Plenge, Nadine Schnepel, Mathias Müsken, Pascal Benz.

**Project administration:** Gemma Mazzuoli-Weber, Pascal Benz.

**Supervision:** Ralph Goethe, Gerhard Breves, Gemma Mazzuoli-Weber, Pascal Benz.

**Visualization:** Masina Plenge, Mathias Müsken.

**Writing – original draft:** Masina Plenge.

**Writing – review & editing:** Nadine Schnepel, Mathias Müsken, Judith Rohde, Ralph Goethe, Gerhard Breves, Gemma Mazzuoli-Weber, Pascal Benz.

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
