## [Decision Letter · Decision Letter 0]

26 Nov 2024

Dear Dr. Plenge,

Thank you for submitting your manuscript to PLOS ONE. After careful consideration, we feel that it has merit but does not fully meet PLOS ONE’s publication criteria as it currently stands. Therefore, we invite you to submit a revised version of the manuscript that addresses the points raised during the review process.

We look forward to receiving your revised manuscript.

Kind regards,

Kevin Looi, Ph.D

Academic Editor

PLOS ONE

Journal Requirements:

“Thanks to the Federal Ministry of Food and Agriculture (BLE# 28N-2-071-00) for funding.”

3. Please note that funding information should not appear in the Acknowledgments section or other areas of your manuscript. We will only publish funding information present in the Funding Statement section of the online submission form. Please remove any funding-related text from the manuscript. 

**Additional Editor Comments:**

Editor's comments to authors:

1) The limited sample size (n=2 pigs) is a major concern and has significant implications on the interpretation of the data. The manuscript should explicitly acknowledge this limitation. Including individual data points in the figures, rather than bar charts, will also improve transparency and help readers better understand data variability. The Methods section, while thorough, contains redundancies that can be streamlined for readability. For instance, routine steps such as RNA extraction need not be described in detail if they follow standard protocols. Clarity is also needed regarding native tissue usage, specifically, whether the samples were derived from the same animals or archived. Additionally, ensure the number of technical replicates is clearly stated, and define all acronyms (e.g., Isc, Rt) upon their first use.

2) The Results section would benefit from adjustments to improve interpretability. For example, Figure 1 should display absolute TEER values rather than percentages to allow for meaningful comparisons with other studies. Similarly, all figures should include individual data points to reflect variability more effectively. Any missing details, such as error bars for ZO-1 gene expression on day 9 in Figure 4, should be added. Enhancing figure legends, such as clarifying aldosterone concentrations in Figure 7, will make the data presentation more complete and accessible. These are examples and the authors should refer to the individual Reviewer comments for further details.

3) A key point raised by the reviewers concerns the lack of response to carbachol in the Ussing chamber experiments, which contrasts with its reported effects in native tissue. If this discrepancy stems from methodological differences, dosing, or tissue-specific factors, these should be thoroughly discussed. While including native tissue in Ussing chamber experiments for direct functional comparison would have been ideal, if this was not feasible, the manuscript should acknowledge and explain this omission.

4) The Discussion section could benefit from a more concise and focused structure. Beginning with a brief summary of the main findings will provide a clear context for readers. Consider removing subheadings and ensuring smooth transitions between topics, such as mucin gene expression, TEER, and other findings. Expanding on functional differences and limitations, including the small sample size, lack of long-term data, and the absence of structural protein analysis will strengthen the discussion and provide valuable context for the study’s conclusions. If data on the long-term stability of the 2D cultures are not available, discussing potential implications and future directions for longer-term studies will still enhance the impact of your work.

Reviewers' comments:

Reviewer's Responses to Questions

**Comments to the Author**

1. Is the manuscript technically sound, and do the data support the conclusions?

Reviewer #1: Yes

Reviewer #2: Partly

2. Has the statistical analysis been performed appropriately and rigorously?

Reviewer #1: Yes

Reviewer #2: Yes

3. Have the authors made all data underlying the findings in their manuscript fully available?

Reviewer #1: No

Reviewer #2: Yes

4. Is the manuscript presented in an intelligible fashion and written in standard English?

Reviewer #1: Yes

Reviewer #2: Yes

Reviewer #1: I enjoyed reading the manuscript by Plenge and colleagues describing the establishment of an air-liquid interphase model of the porcine colonic epithelium. The introduction was well written and adequately presents the study rationale. The methods are appropriate for the study aim. The results are justified throughout the discussion, though the discussion may benefit from some refinement and focus. Overall this is an interesting study and presents a novel approach to allow future studies to experimentally assess the porcine colon without the need for harvesting fresh tissue.

Generalised comments:

1. Perhaps this is naivety as I work primarily with airway epithelium, but it is not clear why stem cells were utilised to construct 3D organoids prior to the 2D culture. Could the stem cells not have been directly differentiated in the ALI culture? The authors may wish to expand the rationale behind this step, as my understanding is that barrier function could still be assessed if proceeding directly to ALI culture.

2. The methods section overall is rather wordy. I think this could be significantly reduced to aid in readability – for example, the RNA extraction paragraph states the kit was performed according to manufacturer's instructions but then proceeds to describe each step.

3. The correct gene notation needs to be utilised throughout the manuscript.

4. I am concerned about the small sample size used for many of the results (typically n=4). I do not think this necessary precludes publication, but I would strongly advocate for the graphs to be updated to include the individual data points so readers can better understand any variation present. The authors should also consider adding a line addressing that this study is a ‘proof of concept’ rather than a detailed comparison against native tissue.

Specific comments:

5. Can the authors please provide additional information regarding the Animal Welfare Act that provides the ethics exemption (year and country).

6. Very minimal information is provided in regard to the native tissue that was utilised. A brief sentence should be included in the methods regarding the RNA collection of native tissue – was it from the same animals or was this banked sample?

7. Lines 76 to 79 repeat the same information provided in lines 70-73.

8. In Figure 1 it would be preferred if the actual TEER values were reported, rather than reporting a percentage. It is difficult to objectively determine whether a robust barrier was formed and compare to other studies when the results have been normalised to a previous timepoint.

9. Similarly, as mentioned earlier, figures 3-8 would significantly benefit from showing the individual data points on the graphs. Some of the error bars appear quite large and given the small number of samples it would help interpretation if individual data points are shown.

10. The arrows in Figure 2 are a bit difficult to see. Could the authors consider enlarging the arrows and perhaps making them less transparent/brighter?

11. There is a minor typo in line 138 – should be epithelial.

12. In line 140 and 141 the abbreviations Isc and Rt are used but a full definition is not provided.

13. The reference or references used in lines 188 and 189 appear to be corrupted/missing.

14. On line 223, the title should be shortened to ‘Ussing chamber studies show physiological transport capacity of 2D-organoids’ as the results do not present any native tissue to assess similarity.

15. The discussion would benefit from a generalised introductory paragraph summarising the main findings. The authors may also wish to remove the subheadings from the discussion, as it could be argued that everything they assessed is considered barrier function.

16. The ‘Barrier Function’ component of the discussion could be streamlined – I can understand what the authors are trying to say and agree with the findings but it jumps between topics with little consistency. For example, Line 272 states “To gain further insight and utilise the model for the study of pathophysiology, an analysis of gene expression was conducted.” Which comes directly after a discussion on mucin gene expression and is immediately followed by discussion of TEER results. Similarly, a strength of the study is the comparison to native tissue, but the mucin similarity is not discussed. Yet it does discuss the comparative expression of the mucins, which one could argue is not particularly relevant if the mucin expression is the same as native tissue.

17. Why do the authors speculate the Ussing chamber is not sensitive to carbachol induced changes when numerous other studies have used Ussing chambers to assess the effects of carbachol? Are they referring to the effect of this tissue specifically? Could it be a dosing issue? I am not familiar with the referenced papers assessing the carbachol response in native tissue, but were there experimental differences that could explain this?

Reviewer #2: The work conducted by Plenge et al in this manuscript characterises a 2D porcine colonic organoid model for studying the physiology and barrier function of the gut. The manuscript is overall well-written, and the techniques used are appropriate. The authors were able to generate a 2D organoid model within 10 days of seeding that recapitulates many of the morphological and transcriptional characteristics observed in the native colonic epithelium, albeit with slight functional differences shown via Ussing chambers. This in vitro model could be utilised for translational research in gastrointestinal physiology, as well as testing intestinal-specific responses to therapeutic applications.

However, my major concern about the study pertains to the extremely limited sample size – the authors only had a biological replicate of two pigs. It is unclear whether the n numbers referenced in the figure legends are technical replicates per animal or the combined total over two animals, or whether multiple samples were taken from each pig. Nevertheless, such a small sample size does not provide the reader with any confidence that the mean (SD) is accurate. Primary cell cultures are known to display larger variations in their morphology and physiology compared to immortalised cell lines. Hence the authors should conduct additional studies to ensure their observations are indeed real and do not occur by chance.

There are a few grammatical and typographical errors that should be amended before acceptance in the journal. The authors are referred to the uploaded feedback where a number of these errors (but not exhaustive) are highlighted. Other general comments are noted below:

1. Lines 75-79 are repeats of lines 70-73. The latter should be deleted.

2. Line 80: Although the procedure has been described in Hoffman et al, the authors should still state which part of the intestine the samples were derived from. Is it also the jejunum? Additional description would be helpful.

3. Line 82: what medium? The reference to S1 table should be made here. Clarify as 'organoid medium' (S1 Table).

4. Line 86: Again, authors are advised to specify which medium - organoid media

5. Line 125-6: genes should be italicised

6. Line 138: unclear how many technical replicates were performed for the Ussing chambers? Is it at least 3?

7. Line 140: First time the acronym is used - should be spelled out here. Short-circuit current (Isc) and resistance (Rt)

8. Line 142: Is Table 1 also the order that the substances were tested?

9. Line 142-3: If all cultures had aldosteroine pre-incubation, the highlighted sentence should immediately follow the first sentence after reference to Hoffman et al.

10. Technically, it would have been nice to see the native tissue also mounted on the Ussing chamber as a true control to the 2D organoids to compare functional physiology.

11. Line 166: what mean values are being compared? Also the same for line 168 – what is the measured variable being compared?

12. Line 172: I disagree with the statement “the number of biological experiments”. Please refer to comments above.

13. Lines 176-178: arguably these sentences could be in the methodology section as it provides background information on the methodological approach.

14. Lines 188-9: Please fix Error Reference source not found

15. Line 190: Why not consider a simple H&E cross-section to complement the data? It would enable the visualization of the different cell types too. Although it is understandable to make assumptions working with monolayers.

16. Do the cultures continue to differentiate past day 10? What happens to the culture if you continue to maintain it?

17. Line 268: please italicise genes.

18. Line 327: Agree with the authors comments. Perhaps this reasoning can be expanded to the junctional proteins too -, it may be useful to observe with immunofluorescence whether the structural orientation of the proteins is the same as in native colon tissue. But limitations are well acknowledged.

FIGURES

ALL figures: please show individual points instead of bar charts for transparency

1. Figure 1: authors are recommended to keep the y-axis origin as 0

2. Figure 3: please show individual points instead of bar charts for transparency. Native is missing an ‘e’ at the end of the legend.

3. Figure 4: as above please show individual points. Missing error bars for day 9 of ZO-1 gene.

4. Figure 7: include aldosterone name on the x-axis for ease of understanding

**Do you want your identity to be public for this peer review?** For information about this choice, including consent withdrawal, please see our Privacy Policy

Reviewer #1: No

Reviewer #2: No

---

## [Author Response · Author response to Decision Letter 1]

7 Jan 2025

Additional Editor Comments:

Editor's comments to authors:

The authors thank the editors for their efforts in summarizing the reviewers' feedback on our manuscript. We have addressed all individual points directly in our responses to the reviewers' comments below.

1) The limited sample size (n=2 pigs) is a major concern and has significant implications on the interpretation of the data. The manuscript should explicitly acknowledge this limitation. Including individual data points in the figures, rather than bar charts, will also improve transparency and help readers better understand data variability. The Methods section, while thorough, contains redundancies that can be streamlined for readability. For instance, routine steps such as RNA extraction need not be described in detail if they follow standard protocols. Clarity is also needed regarding native tissue usage, specifically, whether the samples were derived from the same animals or archived. Additionally, ensure the number of technical replicates is clearly stated, and define all acronyms (e.g., Isc, Rt) upon their first use.

We acknowledge that the use of organoids from two pigs is a limitation and have explicitly addressed this in the manuscript. Each biological replicate corresponds to a separate passage of 3D organoids used to create independent 2D cultures. For gene expression, RNA was isolated from four independent 2D cultures, each with two technical replicates, while transport studies involved four independent experiments with three technical replicates each. These details, along with updated figure legends showing individual data points, provide greater transparency and help readers assess variability. We have streamlined the Methods section by removing redundant details (e.g., RNA extraction protocols) and clarified that native tissue samples were derived from four different animals. Acronyms (e.g., Isc, Rt) are now defined upon first use.

2) The Results section would benefit from adjustments to improve interpretability. For example, Figure 1 should display absolute TEER values rather than percentages to allow for meaningful comparisons with other studies. Similarly, all figures should include individual data points to reflect variability more effectively. Any missing details, such as error bars for ZO-1 gene expression on day 9 in Figure 4, should be added. Enhancing figure legends, such as clarifying aldosterone concentrations in Figure 7, will make the data presentation more complete and accessible. These are examples and the authors should refer to the individual Reviewer comments for further details.

Thank you for your suggestions to enhance the interpretability of the Results section. We have updated all figures to include individual data points to better reflect variability. Regarding Figure 1, we chose to report relative TEER values normalized to day 8 (100%) to emphasize relative changes in barrier integrity across time points. To provide additional clarity, the mean absolute TEER value on day 8 has been included in the figure legend, facilitating comparisons with other studies.

3) A key point raised by the reviewers concerns the lack of response to carbachol in the Ussing chamber experiments, which contrasts with its reported effects in native tissue. If this discrepancy stems from methodological differences, dosing, or tissue-specific factors, these should be thoroughly discussed. While including native tissue in Ussing chamber experiments for direct functional comparison would have been ideal, if this was not feasible, the manuscript should acknowledge and explain this omission.

Thank you for your comment regarding the carbachol response. We acknowledge that the lack of response in our study is not due to Ussing chamber sensitivity, as similar carbachol concentrations have been used successfully in other studies. We believe the discrepancy may be due to post-translational modifications affecting transporter functionality, despite comparable gene expression to in vivo conditions.

Although including native tissue for direct comparison would have been ideal, it was not included in this study. The functional physiology of native tissue in Ussing chambers has been well-established (e.g. . Leonhard-Marek et al. (1), Klinger et al. (3), Bridges et al. (4), Inagaki et al. (5)), so a direct comparison was not necessary for the scope of this work. We have revised the manuscript to clarify these points.

4) The Discussion section could benefit from a more concise and focused structure. Beginning with a brief summary of the main findings will provide a clear context for readers. Consider removing subheadings and ensuring smooth transitions between topics, such as mucin gene expression, TEER, and other findings. Expanding on functional differences and limitations, including the small sample size, lack of long-term data, and the absence of structural protein analysis will strengthen the discussion and provide valuable context for the study’s conclusions. If data on the long-term stability of the 2D cultures are not available, discussing potential implications and future directions for longer-term studies will still enhance the impact of your work.

Thank you for your constructive feedback. We have revised the Discussion section to make it more concise and focused, starting with a brief summary of the main findings to provide clear context for readers. We have also removed subheadings and ensured smoother transitions between topics, such as mucin gene expression, TEER, and other findings. We hope these revisions improve the clarity and impact of the Discussion section.

Reviewers' comments:

5. Review Comments to the Author

Reviewer #1: I enjoyed reading the manuscript by Plenge and colleagues describing the establishment of an air-liquid interphase model of the porcine colonic epithelium. The introduction was well written and adequately presents the study rationale. The methods are appropriate for the study aim. The results are justified throughout the discussion, though the discussion may benefit from some refinement and focus. Overall this is an interesting study and presents a novel approach to allow future studies to experimentally assess the porcine colon without the need for harvesting fresh tissue.

Generalised comments:

1. Perhaps this is naivety as I work primarily with airway epithelium, but it is not clear why stem cells were utilised to construct 3D organoids prior to the 2D culture. Could the stem cells not have been directly differentiated in the ALI culture? The authors may wish to expand the rationale behind this step, as my understanding is that barrier function could still be assessed if proceeding directly to ALI culture.

Thank you for your insightful question. We utilized stem cells to construct the 3D organoids prior to transitioning to the 2D culture. This approach was adopted because 3D organoids contain a higher proportion of stem cells, which are essential for generating the monolayer in the subsequent 2D culture. While direct differentiation of stem cells in the 3D culture is a viable option, it is not be suitable for the transport studies we intend to investigate. Additionally, the use of an air-liquid interface (ALI) culture was not employed in this study, as our focus was specifically on using a submerged 2D culture system to facilitate functional assays, such as electrophysiological measurements in Ussing chambers. ALI cultures are more commonly used to study airway epithelium. Therefore, we have opted to utilise the 2D culture. An alternative approach would be to use an undifferentiated 2D monolayer to form a new 2D monolayer. However, this would be less efficient in terms of both time and cost. The use of 3D organoids provides a more reliable and consistent source of stem cells, which is essential for the formation and differentiation of the monolayer.

2. The methods section overall is rather wordy. I think this could be significantly reduced to aid in readability – for example, the RNA extraction paragraph states the kit was performed according to manufacturer's instructions but then proceeds to describe each step.

Thank you for your comments on the methods section. We have conducted a comprehensive rearranging of the section and agree that certain details could be streamlined to enhance readability. Therefore, the methods section has been significantly shortened to enhance readability (line 112).

3. The correct gene notation needs to be utilised throughout the manuscript.

The gene notation has been verified throughout the manuscript and updated as required.

4. I am concerned about the small sample size used for many of the results (typically n=4). I do not think this necessary precludes publication, but I would strongly advocate for the graphs to be updated to include the individual data points so readers can better understand any variation present. The authors should also consider adding a line addressing that this study is a ‘proof of concept’ rather than a detailed comparison against native tissue.

We are grateful for your considered response. We are aware of the issue of the small sample size and of the necessity to ensure that readers have a comprehensive understanding of the variability in our results. For the gene expression analysis, RNA was isolated from four independent 2D cultures of organoids, with each sample having two technical replicates to ensure the reliability and consistency of the data. Similarly, for the transport characteristics, four independent experiments were conducted, each with three technical replicates, thus facilitating an assessment of the reproducibility of the findings. This has also been included to the manuscript. It is also important to note that the organoids used in this study were derived from two pigs and the 3D organoid cultures were treated with high similarity to an immortalised cell line, thereby facilitating continuous passage and reproducibility due to their high amounts of stem cells. In this context, the passage of the 3D organoids used to generate the 2D cultures can be considered a biological replicate, thereby further reinforcing the robustness of the study. In accordance with your recommendation, we will incorporate individual data points into all the graphs, thus facilitating a more detailed visual representation of the data and any inherent variations.

In regard to the scope of the study, we acknowledge the necessity of explicitly delineating the context within which it is situated. Although this is an early-stage investigation, we have chosen not to frame the work as a proof of concept, as we believe this term can be somewhat limiting and often underestimates the value of studies that explore novel approaches. Instead, our intention is to position this work as a significant contribution towards a deeper understanding of the characteristics of organoid models, with the objective of stimulating further research in this area. We are committed to providing transparency regarding the experimental design, the limitations of the study, and the necessity for further validation in future studies. It is our intention that this approach will provide readers with a clear understanding of the study's intent and its position within the broader context of ongoing research in this field.

Specific comments:

5. Can the authors please provide additional information regarding the Animal Welfare Act that provides the ethics exemption (year and country).

Thank you for your comment. We have included the information regarding the Animal Welfare Act in the document (line 71).

6. Very minimal information is provided in regard to the native tissue that was utilised. A brief sentence should be included in the methods regarding the RNA collection of native tissue – was it from the same animals or was this banked sample?

Thank you for your feedback. The missing information has been included in the method section (line 109-110). The native tissue was sourced from different pigs from banked samples. The pigs were of the same breed and comparable age as the pigs which were used to isolated the crypts.

7. Lines 76 to 79 repeat the same information provided in lines 70-73.

Thank you for bringing this to our attention, the repeated lines have been removed.

8. In Figure 1 it would be preferred if the actual TEER values were reported, rather than reporting a percentage. It is difficult to objectively determine whether a robust barrier was formed and compare to other studies when the results have been normalised to a previous timepoint.

We appreciate your feedback regarding the reporting of TEER values. In the figure, we chose to report the relative TEER values as a percentage normalised to the day 8 as 100 %, as this approach allows for easier comparison of the relative changes in barrier integrity across time points. However, we understand that presenting the actual TEER values could offer additional clarity. To address this, we have included the mean TEER value on day 8 along with the standard deviation in the figure description. We hope this additional information will assist in evaluating the robustness of the barrier formation and facilitates comparisons with other studies.

9. Similarly, as mentioned earlier, figures 3-8 would significantly benefit from showing the individual data points on the graphs. Some of the error bars appear quite large and given the small number of samples it would help interpretation if individual data points are shown.

Thank you for your comment. The figures 3-8 have been changed into discrete data points representing the results of the independent experiments.

10. The arrows in Figure 2 are a bit difficult to see. Could the authors consider enlarging the arrows and perhaps making them less transparent/brighter?

Thank you for bringing this to our attention. The arrowheads have been modified to larger and, hopefully, brighter arrows to improve the visibility to the reader.

11. There is a minor typo in line 138 – should be epithelial.

The typo has been fixed.

12. In line 140 and 141 the abbreviations Isc and Rt are used but a full definition is not provided.

13. The reference or references used in lines 188 and 189 appear to be corrupted/missing.

14. On line 223, the title should be shortened to ‘Ussing chamber studies show physiological transport capacity of 2D-organoids’ as the results do not present any native tissue to assess similarity.

Thank you for your feedback; we have made the requested changes by providing full definitions for Isc and Rt, correcting the references on lines 137.

15. The discussion would benefit from a generalised introductory paragraph summarising the main findings. The authors may also wish to remove the subheadings from the discussion, as it could be argued that everything they assessed is considered barrier function.

We are grateful for your valuable input. In response, we have incorporated a comprehensive introductory paragraph in the discussion section, summarising the primary findings (line 267 – 275). Additionally, we have eliminated the subheadings, aligning with your observation that all assessed elements pertain to the subject of barrier function.

16. The ‘Barrier Function’ component of the discussion could be streamlined – I can understand what the authors are trying to say and agree with the findings but it jumps between topics with little consistency. For example, Line 272 states “To gain further insight and utilise the model for the study of pathophysiology, an analysis of gene expression was conducted.” Which comes directly after a discussion on mucin gene expression and is immediately followed by discussion of TEER results. Similarly, a strength of the study is the comparison to native tissue, but the mucin similarity is not discussed. Yet it does discuss the comparative expression of the mucins, which one could argue is not particularly relevant if the mucin expression is the same as native tissue.

We have ref

---

## [Decision Letter · Decision Letter 1]

21 Feb 2025

Dear Dr. Plenge,

Thank you for submitting your manuscript to PLOS ONE. After careful consideration, we feel that it has merit but does not fully meet PLOS ONE’s publication criteria as it currently stands. Therefore, we invite you to submit a revised version of the manuscript that addresses the points raised during the review process.

We look forward to receiving your revised manuscript.

Kind regards,

Kevin Looi, Ph.D

Academic Editor

PLOS ONE

Journal Requirements:

Reviewers' comments:

Reviewer's Responses to Questions

**Comments to the Author**

Reviewer #1: (No Response)

Reviewer #2: (No Response)

2. Is the manuscript technically sound, and do the data support the conclusions?

Reviewer #1: Yes

Reviewer #2: Yes

3. Has the statistical analysis been performed appropriately and rigorously?

Reviewer #1: Yes

Reviewer #2: Yes

4. Have the authors made all data underlying the findings in their manuscript fully available?

Reviewer #1: Yes

Reviewer #2: Yes

5. Is the manuscript presented in an intelligible fashion and written in standard English?

Reviewer #1: Yes

Reviewer #2: Yes

Reviewer #1: The authors have done well to address the initial reviewer comments. I am satisfied with the science but have some minor comments to improve the manuscript:

The description of the native tissue has been added as requested; however, it is described without any prior introduction. For reader clarity, this should be slightly modified to make clear that native tissue was used for a comparator before they give the details of where the tissue was collected.

The discussion would benefit from addressing the small sample size – it was rationalized in the reviewer responses but is not specifically addressed in the manuscript. The inclusion of specific sample sizes was an important and welcome addition. However, it is still a small sample and that does limit the study findings. Given many readers will be critical of the work because of the limited sample number, the authors should pre-emptively address these concerns.

Line 283: Space missing between mucins and were

Line 314: Typographical error in ‘expression’.

Line 330: Typographical error in ‘replicated’.

Reviewer #2: Thank you to the authors for addressing many of the comments from the first review. The manuscript has improved clarity and transparency, with apt acknowledgment of model limitations and future investigations. This novel culture model will enable investigations into the pathophysiology of intestinal diseases, which are increasingly becoming prevalent.

Minor comments - Thank you for changing the figures to represent individual data points. However, error bars are missing from figures 3-8 (despite the figure legend indicating mean+- SD). While the individual datapoints allow readers to deduce the spread of the data, it would still be nice to have the T error bars to indicate the calculated SD. This can be added behind the datapoints.

**Do you want your identity to be public for this peer review?** For information about this choice, including consent withdrawal, please see our Privacy Policy

Reviewer #1: No

Reviewer #2: No

---

## [Author Response · Author response to Decision Letter 2]

25 Feb 2025

Review Comments to the Author

Reviewer #1: The authors have done well to address the initial reviewer comments. I am satisfied with the science but have some minor comments to improve the manuscript:

The description of the native tissue has been added as requested; however, it is described without any prior introduction. For reader clarity, this should be slightly modified to make clear that native tissue was used for a comparator before they give the details of where the tissue was collected.

Thank you for your constructive comments on our manuscript. We have revised the methods section to introduce the native tissue as a comparator, providing an introduction.

The discussion would benefit from addressing the small sample size – it was rationalized in the reviewer responses but is not specifically addressed in the manuscript. The inclusion of specific sample sizes was an important and welcome addition. However, it is still a small sample and that does limit the study findings. Given many readers will be critical of the work because of the limited sample number, the authors should pre-emptively address these concerns.

Thank you for pointing this out. We have now addressed the small sample size directly in the manuscript. This should address concern from readers regarding this limitation.

Line 283: Space missing between mucins and were

Line 314: Typographical error in ‘expression’.

Line 330: Typographical error in ‘replicated’.

The typographical errors have been corrected.

Reviewer #2: Thank you to the authors for addressing many of the comments from the first review. The manuscript has improved clarity and transparency, with apt acknowledgment of model limitations and future investigations. This novel culture model will enable investigations into the pathophysiology of intestinal diseases, which are increasingly becoming prevalent.

Minor comments - Thank you for changing the figures to represent individual data points. However, error bars are missing from figures 3-8 (despite the figure legend indicating mean+- SD). While the individual datapoints allow readers to deduce the spread of the data, it would still be nice to have the T error bars to indicate the calculated SD. This can be added behind the datapoints.

Thank you for your positive feedback. We also appreciated your suggestion regarding the error bars. They have been added to the figures 3-8.

---

## [Decision Letter · Decision Letter 2]

14 Mar 2025

Dear Dr. Plenge,

Thank you for submitting your manuscript to PLOS ONE. After careful consideration, we feel that it has merit but does not fully meet PLOS ONE’s publication criteria as it currently stands. Therefore, we invite you to submit a revised version of the manuscript that addresses the points raised during the review process.

Please submit your clarification and comments by Apr 28 2025 11:59PM. If you will need more time to supply your clarifications and comments, please reply to this message or contact the journal office at plosone@plos.org . A rebuttal letter that responds to each point raised by the academic editor and reviewer(s). You should upload this letter as a separate file labeled 'Response to Reviewers'.A marked-up copy of your manuscript that highlights changes made to the original version. You should upload this as a separate file labeled 'Revised Manuscript with Track Changes'.An unmarked version of your revised paper without tracked changes. You should upload this as a separate file labeled 'Manuscript'.

We look forward to receiving your revised manuscript.

Kind regards,

Kevin Looi, Ph.D

Academic Editor

PLOS ONE

Additional Editor Comments :

Dear Dr Plenge,

We are currently reviewing your manuscript, titled “Development and Characterization of a 2D Porcine Colonic Organoid Model for Studying Intestinal Physiology and Barrier Function”.

During our assessment, we identified a highly similar manuscript published in Journal of Visualized Experiments titled “Two-dimensional Porcine Intestinal Organoids Reflecting the Physiological Properties of Native Gut, DOI: 10.3791/67666.” The significant overlap in content between the two manuscripts raises concerns regarding potential duplication of publication.

To ensure transparency and adherence to ethical publishing standards, we kindly request clarification on the following points:

1) The relationship between the two manuscripts (e.g., whether they represent distinct studies or overlapping analyses).

2) Whether the submission to Journal of Visualized Experiments was made with the understanding that the content had already been accepted/published elsewhere.

3) Any additional context or justification for the similarities between the manuscripts.

We take issues of publication ethics very seriously and aim to resolve this matter promptly. Your response will help us better understand the situation and determine the appropriate next steps.

Please provide your clarification at your earliest convenience. If you have any questions or require further assistance, do not hesitate to reach out to the PLOS One team.

Thank you for your attention to this matter, and we look forward to your response.

Reviewers' comments:

Reviewer's Responses to Questions

**Comments to the Author**

Reviewer #1: All comments have been addressed

Reviewer #2: All comments have been addressed

2. Is the manuscript technically sound, and do the data support the conclusions?

Reviewer #1: Yes

Reviewer #2: Yes

3. Has the statistical analysis been performed appropriately and rigorously?

Reviewer #1: Yes

Reviewer #2: Yes

4. Have the authors made all data underlying the findings in their manuscript fully available?

Reviewer #1: Yes

Reviewer #2: Yes

5. Is the manuscript presented in an intelligible fashion and written in standard English?

Reviewer #1: Yes

Reviewer #2: Yes

Reviewer #1: (No Response)

Reviewer #2: A final minor correction - I noticed that in Figure 3, 2nd panel (relative gene expression of MUC genes) the y-axis changed to -1 (compared to Revision 1). The y-axis origin should be 0, consistent with other panels. Please update figure 3. Otherwise, the authors have addressed all concerns.

**Do you want your identity to be public for this peer review?** For information about this choice, including consent withdrawal, please see our Privacy Policy

Reviewer #1: No

Reviewer #2: No

---

## [Author Response · Author response to Decision Letter 3]

19 Mar 2025

Dear Editor,

Thank you for your detailed feedback regarding the perceived overlap between our revised manuscript and our previously published work in the Journal of Visualized Experiments (DOI: 10.3791/67666). We appreciate the opportunity to clarify the relationship between these two studies.

1. Independence of the studies

Although both manuscripts use the two-dimensional porcine intestinal organoid model, they are based on independent experiments with different scientific objectives. The JOVE paper focuses primarily on establishing and visually demonstrating the protocol for generating 2D organoid cultures, thus providing a validated methodological framework. In contrast, our revised manuscript presents a detailed functional and molecular characterisation of the model, with particular emphasis on intestinal barrier integrity, ion transport and gene expression.

2. Experimental use of different donor animals and timeline

The two studies were performed using organoids derived from different donor animals, as reflected by the different animal registration numbers reported in the respective manuscripts (e.g. JOVE: TiHo-T-2023-15; revised manuscript: TiHo-T-2017-22 and TiHo-T-2024-5). In addition, the experiments were carried out in different time periods, further confirming their independence. Despite these differences, the high similarity of the results obtained in both studies underlines the robustness and reproducibility of the 2D porcine organoid model.

3. Justification for Methodological Overlap

As both studies use the same well-established organoid system, it is inevitable that certain parts of the materials and methods will be similar. However, the research questions, experimental approaches and data analyses are distinctly different between the two manuscripts. The JOVE study serves as a methodological reference, while the revised manuscript extends this work by exploring unique aspects of the model that are critical for understanding its physiological and pathophysiological properties.

In summary, although both manuscripts share a common experimental platform, they represent independent investigations with complementary but non-duplicative aims. We hope that this clarification reinforces our commitment to transparency and adherence to ethical publishing standards.

Sincerely,

Masina Plenge

---

## [Decision Letter · Decision Letter 3]

4 Apr 2025

Development and Characterization of a 2D Porcine Colonic Organoid Model for Studying Intestinal Physiology and Barrier Function

PONE-D-24-46719R3

Dear Dr. Plenge,

We’re pleased to inform you that your manuscript has been judged scientifically suitable for publication and will be formally accepted for publication once it meets all outstanding technical requirements.

Kind regards,

Mária A. Deli, M.D., Ph.D.

Academic Editor

PLOS ONE

Additional Editor Comments (optional):

Reviewers' comments:

Reviewer's Responses to Questions

**Comments to the Author**

Reviewer #1: All comments have been addressed

Reviewer #2: All comments have been addressed

2. Is the manuscript technically sound, and do the data support the conclusions?

Reviewer #1: Yes

Reviewer #2: Yes

3. Has the statistical analysis been performed appropriately and rigorously?

Reviewer #1: Yes

Reviewer #2: Yes

4. Have the authors made all data underlying the findings in their manuscript fully available?

Reviewer #1: Yes

Reviewer #2: Yes

5. Is the manuscript presented in an intelligible fashion and written in standard English?

Reviewer #1: Yes

Reviewer #2: Yes

Reviewer #1: (No Response)

Reviewer #2: Thanks to the authors for clarifying. I recognise that the two studies were completed during an independent time period and do not share the same animal registration numbers. The JoVe article describes the methodologic aspects of monolayer formation whereas the current revised manuscript explores the physiological characteristics.

line 348 - missing a space between "strong" and "foundation". This can be edited during typesetting.

**Do you want your identity to be public for this peer review?** For information about this choice, including consent withdrawal, please see our Privacy Policy

Reviewer #1: No

Reviewer #2: No

---

## [Editor Report · Acceptance letter]

PONE-D-24-46719R3

PLOS ONE

Dear Dr. Plenge,

I'm pleased to inform you that your manuscript has been deemed suitable for publication in PLOS ONE. Congratulations! Your manuscript is now being handed over to our production team.

Kind regards,

on behalf of

Prof. Mária A. Deli

Academic Editor

PLOS ONE